# Unravelling the Roles of Nitrogen Nutrition in Plant Disease Defences

**DOI:** 10.3390/ijms21020572

**Published:** 2020-01-16

**Authors:** Yuming Sun, Min Wang, Luis Alejandro Jose Mur, Qirong Shen, Shiwei Guo

**Affiliations:** 1Jiangsu Provincial Key Lab of Solid Organic Waste Utilization, Jiangsu Collaborative Innovation Center of Solid Organic Wastes, Educational Ministry Engineering Center of Resource-saving fertilizers, Nanjing Agricultural University, Nanjing 210095, China; sunyumingagw@163.com (Y.S.); shenqirong@njau.edu.cn (Q.S.); sguo@njau.edu.cn (S.G.); 2Jiangsu Key Laboratory for the Research and Utilization of Plant Resources, Institute of Botany, Jiangsu Province and Chinese Academy of Sciences, Nanjing 210014, China; 3Institute of Biological, Environmental and Rural Sciences, Aberystwyth University, Aberystwyth SY23 3DA, UK; lum@aber.ac.uk

**Keywords:** nitrogen, plant disease, physical, biochemical, molecular, disease defence

## Abstract

Nitrogen (N) is one of the most important elements that has a central impact on plant growth and yield. N is also widely involved in plant stress responses, but its roles in host-pathogen interactions are complex as each affects the other. In this review, we summarize the relationship between N nutrition and plant disease and stress its importance for both host and pathogen. From the perspective of the pathogen, we describe how N can affect the pathogen’s infection strategy, whether necrotrophic or biotrophic. N can influence the deployment of virulence factors such as type III secretion systems in bacterial pathogen or contribute nutrients such as gamma-aminobutyric acid to the invader. Considering the host, the association between N nutrition and plant defence is considered in terms of physical, biochemical and genetic mechanisms. Generally, N has negative effects on physical defences and the production of anti-microbial phytoalexins but positive effects on defence-related enzymes and proteins to affect local defence as well as systemic resistance. N nutrition can also influence defence via amino acid metabolism and hormone production to affect downstream defence-related gene expression via transcriptional regulation and nitric oxide (NO) production, which represents a direct link with N. Although the critical role of N nutrition in plant defences is stressed in this review, further work is urgently needed to provide a comprehensive understanding of how opposing virulence and defence mechanisms are influenced by interacting networks.

## 1. Introduction

Throughout its life, a plant is often exposed to a wide range of soil-borne or air-borne pathogens that represent a threat to plant normal growth and seed production. Therefore, pathogens can have considerable impacts on agricultural productivity and with major economic consequences. With millions of years of evolution, plants have formed multi-layered defence systems to perceive and resist invasion by these various types of pathogenic microorganisms, which include bacteria, fungi, nematodes or viruses. Simply classified, the plant defence systems can be divided into constitutive and induced modes [1]. Constitutive defence tends to involve only the outmost layer of plant tissues and, therefore, tends to be the first defensive line to be exposed to the invading pathogens. Typically, this form of defence consists of physical barriers and pre-formed chemical compounds (Figure 1) [2,3]. The strength of the physical barrier directly determines the number of pathogens entering the plant so that changes in the cuticle permeability, cell wall thickness and the degree of lignification critically affect plant resistance to pathogen infection [4,5]. The associated chemical defences consist primarily of secondary metabolites that inhibit microbial growth and further invasion. Antimicrobial phytochemicals may also serve as defence-related signallings to augment their importance on plant immunity [6,7].

Plants can also recognize pathogen (or microbial)-associated molecular patterns (PAMPs/MAMPs) or damage-associated molecular patterns (DAMPs) through pattern recognition receptors (PRRs) to induce PAMP-triggered immunity (PTI) [8,9]. However, pathogens can evolve effectors to suppress PTI, leading to effector-triggered susceptibility (ETS). If effectors are recognized by Resistance (*R*) gene encoded nucleotide-binding leucine-rich repeat (NB-LRR) proteins, plants can deploy effector-triggered immunity (ETI) [10]. ETI is a more prolonged and robust response than PTI but can only be deployed against specific pathogen infections [11]. Both PTI and ETI act through integrated signalling networks to defend against pathogens in a more specific manner with varying responses depending on the interacting pathogen. The outcomes of PTI/ETI can include the expression of differing defence genes, the formation of phytoalexins and the development of a hypersensitive response (HR) at the infection site [9,12]. Such defence responses can be reiterated throughout the plant to confer systemic acquired resistance (SAR). These induced defences can also be elicited by beneficial microbes, herbivores, chemical inducer or abiotic stress [13]. The pathogen-induced SAR occurs in the distal tissue to enhance the broad-spectrum resistance against secondary related or unrelated pathogenic challenges. Such resistance is achieved via the action of signalling compounds generated at the site of primary infection, such as reactive oxygen species (ROS), nitric oxide (NO) and salicylic acid (SA) (Figure 1) [14,15].

The interaction between plants and pathogenic microorganisms is therefore complex but can be influenced by multiple environmental factors such as temperature, humidity, light and nutrients. In this context, nitrogen (N) is particularly important as an essential macroelement for plant normal growth and development, which accounts for ~1.5% to 2% of plant dry matter and ~16% of total plant proteins [16]. N assimilation is related to key physiological or metabolic processes such as photosynthesis, photorespiration, respiration, amino acid synthesis and the tricarboxylic acid (TCA) cycle [17,18]. Given its centrality, it is unsurprising that a plant’s N status can influence plant resistance to varied abiotic and biotic stresses. For instance, numerous studies have shown that N supply can alter plant resilience to abiotic stresses possibly through effect-associated plant growth patterns and N-mediated signalling transduction [19]. In the context of plant disease, N availability can limit pathogen growth and affect the elicitation and deployment of plant defences. Additionally, the supply of different forms of N (ammonium [NH_4_^+^] vs. nitrate [NO_3_^−^]) appears to have different effects on plant disease resistance, at least in part through the use of different assimilation and metabolism pathways [20,21]. However, such studies are primarily observational and descriptive, and the underlying defence mechanisms mediated by N are still not fully understood. Further, the impact on N of both of the interacting partners, host and pathogen, needs to be more clearly defined. In this review, we comprehensively consider the relationship between N nutrition and plant disease incidence as suggested by its effects on host and pathogen from physical, biochemical and molecular perspectives. 

## 2. N Nutrition as It Impacts on the Pathogen

There are many instances where N fertilization can increase plant disease incidence, for example, with downy mildew, powdery mildew, leaf rust, stem rot and rice blast diseases [22,23,24,25]. Equally, the opposite effects have been reported for take-all, grey mould and leaf spot disease [26,27,28,29] (Table 1). These different outcomes cannot be associated with a given host (e.g., wheat) or different infection strategy, whether necrotrophic or (hemi)biotrophic or if a foliar or root-infecting pathogen. Necrotrophic pathogens cause host cell death through the production of toxins or enzymes, whilst biotrophs maintain host cell viability for an extended period. With time, biotrophs can switch to a necrotrophic mode of pathogenesis and are hence designated hemibiotrophs. Previous studies indicated that N fertilization could mostly increase plant disease incidence with biotrophic pathogen infection, while the opposite result occurs when challenged with necrotrophs such as *Alternaria* and *Fusarium* [30,31]. Whilst this could indicate that the requirement for viable host cells makes biotrophs sensitive to host N status, in *Arabidopsis* resistance to necrotrophic pathogens, *Erwinia amylovora* and *Botrytis cinerea* are inversely regulated by N nutrition [31,32]. Instead, it seems likely to reflect the interplay of specific elicitation events between pathogen and host. These elicitation events need to be defined and they could be used to improve crop yield without compromising resistance to disease.

In addition, factors such as plant growth stages, cultivars and planting years would also induce conflicting effects on the correlation between N nutrition and the effectiveness of pathogen infection [33,34]. All these discrete influences most likely reflect either the requirement of pathogenic proliferation or virulence on the availably of N [20,30]. As an additional complication, the form of N fertilization can change plant disease severity. Thus, NH_4_^+^ nutrition significantly increased plant resistance to take-all disease in wheat [35], summer patch in bluegrass [36] and black root rot in strawberry [37], while NO_3_^−^ nutrition induced plant resistance to *Fusarium* infection [38,39].

From the perspective of pathogens, N-promoted plant growth provides increased succulent tissues, apoplastic amino acid concentrations and improved plant canopy structure, which would all favour the growth of pathogenic spores [40,41]. On infection, pathogens require a wide range of N sources, including NH_4_^+^ and NO_3_^−^ as well as amino acids. Studies have shown that biotrophic pathogen infection induced amino acid accumulation at the infection site, among which gamma-aminobutyric acid (GABA) is an important N source for the development of *Cladosporium fulvum* [42], *Stagonospora nodorum* and *Fusarium graminearum* [43]. Additionally, different forms of N nutrition affect the development of pathogenic microorganisms. Unlike NO_3_^−^ feeding, NH_4_^+^ can inhibit the proliferation of *Fusarium oxysporum* [44,45], *Vetticillium dahlia* [46] and *Elsinochrome* [47], while the opposite events were also reported in *Fusarium oxysporum* f. sp. *citri* [38], *aflatoxin* [48] and various root rot diseases [49].

Direct N impacts on pathogen virulence have been noted with effector delivery. Thus, N starvation stimulates pathogen effector genes, such as the *hrp* (hypersensitive response and pathogenicity), *Avr* (avirulence) and hydrophobin *MPG1* genes in *Magnaporthe oryzae* [50,51]. In an extensive transcriptomic assessment of the impact of N limitation on *Pseudomonas syringae* pv*. syringae* B728a, it was suggested that virulence-associated features such as swarming motility, type III secretion and metabolic pathways involved in GABA and polyketide metabolism were prominent [21]. Such studies indicate the importance of N-starvation in initiating pathogenesis. Equally, the opposite effects have been documented for effectors from *Magnaporthe oryzae* [22] and *Cladosporium fulvum* [52]. However, the changes of pathogens alone cannot explain the discrete relationship between N nutrition and plant disease incidence. It becomes more complex when understanding it from the perspective of plants, as the effect of N nutrition on plant defence mechanisms.

## 3. N Nutrition as It Impacts on Host Defence

### 3.1. Physical Defence Mechanisms

Plant N availability modulates cellular structure and composition via its effects on plant in vivo primary and secondary metabolism. These will affect plant disease defences by affecting the thickness of the plant’s physical barrier. Generally, increased N input will promote plant growth but at the price of less formation of lignin and waxy cuticle. The negative correlation between plant N status and surface wax density was demonstrated in both *Pinus Palustris* and Norway spruce seedlings [53,54]. Further, use of histochemical staining, biochemical assays and gene expression patterns have consistently demonstrated the negative impact of increased N availability on the degree of lignification of woody plant tissues [55,56]. Delayed lignin deposition on the xylem cell wall was observed when plants are exposed to excess N input [56]. In another study, high N fertilization resulted in a reduction in the thickness of the secondary cell wall as well as major biopolymer components (cellulose and lignin) in two japonica rice cultivars [57]. Interestingly, agronomically reducing the N fertilization rate can effectively increase plant lodging resistance, which is associated with changes in stem lignification and secondary cell wall synthesis and also emphasizes the negative correlation between N availability and plant epidermal hardness [58,59].

If these structures are compromised, plants will become conducive to penetration by pathogenic microorganisms as well as herbivorous insects [60]. This could partially explain the differences in plant susceptibility to various diseases influenced by N nutrition that are associated with physical resistance. Negative effects of N fertilization on epidermal cuticle thickness as well as disease susceptibility have been seen in the leaf of tomato [61] and fruit of peach [62]. The degree of plant epidermal lignification can effectively prevent the penetration of pathogens, thereby reducing the incidence of disease [4]. Furthermore, as a key enzyme in lignin biosynthesis, cinnamyl alcohol dehydrogenase is negatively regulated by N fertilization, and its inhibition makes *Arabidopsis* more susceptible to *Pseudomonas syringae* pv. *tomato* [63]. However, Nobuaki [64] reported the contradictory impacts of N fertilization on leaf cell wall thickness and blast disease in rice plants, although this could represent a requirement for particular cuticular characteristics to allow host penetration by *Magnaporthe grisea* [65]. In addition, different forms of N nutrition can also influence the strength of the plant’s physical defence. For instance, compared with NO_3_^−^, NH_4_^+^-based N nutrition significantly increased the amount of epicuticular wax of kohlrabi leaves [66]. Wang et al. [67] showed that NH_4_^+^ increases the activities of four peroxidase (POD) isoenzymes involved in lignin synthesis.

To further explain the regulation mechanism of N nutrition on physical defence compounds, the growth-defence balance hypothesis (GDBH) or carbon (C)-nutrient balance hypothesis (CNBH) could be informative. These emphasize either the balance between plant growth and defence or the importance of C/N ratio in C-based secondary metabolism [68,69]. These two theories are not mutually exclusive and would explain changes in C-base secondary metabolites such as lignin and cellulose. Thus, increased N supply promotes primary metabolism to drive growth, which inevitably inhibited C-base secondary metabolism [70]. This in turn would support the hypothesis of a negative relationship between N nutrition and physical defence strength.

### 3.2. Plant Biochemical Defence Mechanism

Biochemical defences are mainly achieved through multiple defence-related enzymes acting to affect primary or secondary metabolites. N mediated biochemical defences are associated with (1) plant metabolites including phytoalexins, antimicrobial proteins, amino acids and organic acids and (2) defence-related enzymes [71,72,73].

#### 3.2.1. Plant Metabolites and Biochemical Defence

The class of metabolites known as phytoalexins are important components against pathogens. Phytoalexins are non-specific antimicrobial metabolites of low molecular weight that can effectively inhibit mycelial growth and spore germination. A reduction of phytoalexins contents increases plant susceptibility to disease [74]. Phytoalexins exhibit enormous chemical diversity, with the most studied being the flavonoids in legume (e.g., isoflavones, coumarin and chlorogenic acid), terpenoids in Solanaceae and Convolvulaceae (e.g., gossypol, kauralexins and zealexins), stilbenes (e.g., resveratrol) and indoles (e.g., camalexin, brassinin, glucosinolates and brassilexin) [75,76,77].

Since most of the phytoalexins are C-based secondary metabolites, their contents are also likely to be negatively regulated by N availability as predicted by the CNBH or GDBH (Table 2). For example, tests in soybean, *Salix polaris* and subarctic tundra heath vegetation showed a negative regulation of N on plant defensive compounds such as coumestrol, total phenols or condensed tannins [78,79,80]. In another study, fungistatic phenolic compounds such as (−)-epicatechin and piceatannol in beech roots and 4-hydroxyacetophenone and piceatannol in Norway spruce root also decreased with increasing N fertilization; however, there was an increase in protocatechuic acid content [81]. These findings suggest that N addition generally reduces the amount of defensive compounds and affects plant disease resistance. Thus, N addition compromised rice resistance to sheath rot (*Sarocladium oryzae*) [82], grapevines to powdery mildew (*Uncinula necator*) [83], *Medicago truncatula* to *Aphanomyces euteiches* [84], potato to *Phytophthora infestans* [85] and apple tree to *Venturia inaequalis* [86]. However, N forms also affect the synthesis of C-based secondary metabolites. For instance, Wu et al. [87] found that NO_3_^−^ nutrition increased total phenol and flavonoids in *Echinacea angustifolia* and total flavonoids in *Hypericum perforatum* [88] and in Leafy Brassica [89]. Similarly, higher NO_3_^−^/NH_4_^+^ ratios in Murashige and Skoog medium increased the production of C-based secondary metabolites, including ginseng saponin, artemisinin and periplocin [90,91,92]. The promoting effect of NO_3_^−^ on secondary metabolites, including various phytoalexins, may be linked to defence simulated by nitrate reductase (Nr)-derived NO signals (considered below).

Most of the antimicrobial proteins are N-based compounds and their responses to N availability are different from those of phenolic-defensive compounds (Table 2). Pathogenesis-related (PR) proteins, chitinase and β-glucanase can degrade the glycoside peptide of bacterial cell wall and chitin or dextran of the fungal cell wall to counter pathogen infection [99]. Thus, the inhibition or induction of PR proteins will result in either a decreased or enhanced plant resistance [100]. The activity of chitinase (PR3), as well as β-1,3-glucanase (PR-2 protein) and chitosanase, increased with the increased N fertilization rate [1,93], which correlated with enhanced plant disease resistance. Correspondingly, significant down-regulation of genes encoding class IV chitinase (3.7-fold) and class III chitinase (10.2-fold) was observed in rice when exposed to low N stress [101]. Another study documented the increased expression of PR4 with increased N [102]. These findings indicate a positive effect of N nutrition on the expression of PR proteins which could be linked to disease resistance. However, no significant variation in chitinase activities was observed in *Arabidopsis thaliana* exposed to different N rates [103]. Another important class of antifungal proteins, protease inhibitors, also play a role in plant immunity [104]. These exhibit positive [105], negative [106] and non-significant [107] effects following N manipulation. Further, any positive effects of N on PR gene expression would appear to be countered by wider negative impacts on defence in particular interactions. A proteomic study on rice-*Magnaporthe grisea* showed that the expression of a sulfur-rich thaumatin-like protein (PR-5) (an antimicrobial protein) was enhanced following challenge with blast fungus incubation under high N feeding [95], although high N made rice susceptible to blast disease. Thus, the N mediated changes in PR proteins are not necessarily consistent with disease resistance, which arises from multiple factors.

Primary C–N metabolites, mainly amino acids and organic acids, link N nutrition and disease due to the need of the pathogen to derive nutrients from the host [108]. Amino acid and organic acid metabolism can be markedly affected by either the rate of N application or its forms. N input generally enhances amino acid content but restricts organic acid metabolism [109]. NH_4_^+^ nutrition increased amino acid content of cucumber plants when compared with NO_3_^−^ nutrition, while the opposite results were exhibited for organic acids [71,97].

Amino acid metabolism is disrupted during plant disease, affecting the overall accumulation patterns or those of individual amino acids. This is particularly relevant as recent functional studies in Arabidopsis indicate that amino acid metabolism contributes more to plant disease defence than phytoalexins [72,110]. Furthermore, metabolically, amino acids are in fact precursors of many defence-related phytoalexins. For example, glucosinolate is a major phytoalexin in Brassica plants, and different categories of glucosinolates are derived from amino acids like alanine, phenylalanine and tryptophan [111]. Equally, high N-induced glucosinolate accumulation has also been reported in Brassica plants [112]. The acylation of amino acids can also affect plant resistance to microbial pathogens and insects. N-acetylornithine derived from arginine, proline and glutamate catabolic pathways increases the susceptibility of Arabidopsis to *Pseudomonas syringae* infection but protects plants from feeding by aphid and herbivores [72,113].

Predictably, the direct manipulation of amino acids can increase disease resistance. Exogenous treatment of proline could enhance the expression of *PR* as well as HR-mediated programmed cell death (PCD), with similar results elucidated in homoserine- or threonine-mediated plant immunity to oomycete pathogens [72,114]. Deletion or overexpression of *asparagine synthetase 1* (*CaAS1*) respectively decreased or enhanced *Arabidopsis thaliana* defence to *Pseudomonas syringae* pv. *tomato* DC3000 and *Hyaloperonospora arabidopsidis* [115], whilst overexpression of *aspartate aminotransferase* resulted in increased plant disease incidence in *Arabidopsis thaliana* challenged with *Botrytis cinerea* [116]. Liu et al. [117] also demonstrated that the knockout of amino acid transporters such as LYSINE HISTIDINE TRANSPORTER1 (LHT1) would reduce the contents of glutamine, alanine, and proline but promote callose deposition and HR, thereby increasing Arabidopsis disease resistance. Beyond amino acids, wider amide metabolism appears to be affected by N supply. The oxidation of polyamines generates extracellular hydrogen peroxide (H_2_O_2_) to trigger PCD or basal resistance and thus enhance plant disease resistance. A positive relationship between polyamine content and N supply levels has also been demonstrated [118,119].

When it comes to the link between organic acids and plant immunity, the most well-characterized is the link between phenolic acids acting as phytoalexins, which are negatively controlled by N nutrition (as discussed above). However, other organic acids can also interfere with plant defence by affecting key metabolic processes. Thus, organic acids involved in the TCA cycle, including oxalic acid, malic acid, citrate acid, succinic acid and fumaric acid, can significantly affect plant resistance, as shown in the cucumber-Fusarium interaction when supplied with NO_3_^−^ [39,71]. In addition, organic acids involved in photorespiration such as glycolic acid and glyoxylic acid also contribute to plant disease resistance. Inhibition of the glycolate oxidase, a component of the photorespiration cycle, compromised disease resistance and such processes are positively regulated by N nutrition, especially in the forms of NO_3_^−^ [120,121]. Organic acids can also participate in plant defence as they include some signalling molecules. These include the important defence hormones SA and jasmonic acid (JA), which form regulatory networks with ethylene (ETH) and could be regulated by N nutrition (discuss below).

#### 3.2.2. Defence-Related Enzymes

Activation of plant defence-related enzymes is another facet in fighting against pathogen invasion, and N is involved in the stimulation of these enzymes during plant-pathogen interactions (Table 2) [73,96]. The phenylpropanoid pathway plays an important part in defence and the key regulatory enzyme in this pathway, phenylalanine ammonia_lyase (PAL), is involved in the synthesis of secondary antimicrobial compounds [122]. Generally, genes encoding general phenylpropanoid metabolism such as *PAL*, *cinnamate-4-hydroxylase* (*C4H*) and *4-coumarate: CoA ligase* (*4CL*) are all up-regulated by N deficiency [56,123], while decreased PAL activity has been seen with N fertilization [60,124]. However, discrete responses were reported when challenged with pathogens. For example, Thapa et al. [96] reported inhibited PAL activity as well as decreased plant resistance by N input in rice-*Magnaporthe* relations. A similar result was also observed in *Medicago truncatula* resistance to *Aphanomyces euteiches* [84]. In contrast, Jin et al. [93] demonstrated increased PAL activity and plant disease resistance by N fertilization, while Ballini et al. [24] reported the upregulation of genes encoding PAL and POD together with increased disease incidence due to high N supply. Furthermore, as a key enzyme converting phenolic compounds into antimicrobial hydrazines, polyphenol oxidase (PPO) also has a role in plant disease resistance; however, the effects of N fertilization had both positive [93] and negative [125] effects. These different responses suggest that the activation of these defence-related enzymes may not be the decisive strategy in N-mediated plant defence or have different roles in differing interactions.

In addition, N nutrition also influences plant antioxidant systems, which play key roles in plant defence responses (Figure 1, Table 2). Antioxidant enzymes include superoxide dismutase (SOD), catalase (CAT), ascorbate peroxidase (APX), glutathione reductase (GR) and POD. These antioxidant enzymes reduce ROS under stresses to less toxic compounds [126]. N fertilization generally increases the activities of these enzymes, thereby enhancing plant antioxidant capacity and reducing cell membrane injury [127,128]. The N deficient-triggered H_2_O_2_ or superoxide anion accumulation in rice leaves [129], *Matricaria chamomilla* roots [130] and wheat peduncles [131] further support these observations. When considering the impact of different N sources, enhanced antioxidant enzyme activities and antioxidants (reduced glutathione, GSH and ascorbate, ASA) contents together with the decreased ROS levels were observed under NO_3_^−^ feeding, especially after exposing to stress [132,133], which may be again associated with the NO-triggered defence responses [134]. However, antioxidant systems may be sensitive to NH_4_^+^ feeding, which may again be associated with the oxidative damage due to “ammonium toxicity” [67,135]. Taking all of these points together, it seems that N can positively participate in plant biochemistry defence against biotic stresses via the antioxidant system.

### 3.3. Plant MOLECULAR Defence Mechanism

#### 3.3.1. N Metabolism Links Hormones and Nitric Oxide Impacts on Defence

Plant hormones play critical roles in plant growth and development, among which SA, JA and ETH are essential regulators of plant defence responses [136]. SA signalling is mainly deployed when plants are challenged with biotrophic and hemibiotrophic pathogens, while JA and ET act synergistically against necrotrophic pathogens. A negative effect of N nutrition on SA accumulation has been reported in *Arabidopsis* leaves, but the response under disease was not tested [137]. The plant’s in vivo N status can also influence the SA levels [137] or defence gene expression (e.g., *PR2* and *WRKY33*) [138] via the action of RING-type ubiquitin E3 ligase. Interestingly, SA-mediated plant resistance is also linked to amino acids. For instance, SAR was induced in rice to *Magnaporthe oryzae* by amino acids, and the broad-spectrum local defence also linked to SA signalling [117]. The JA/ ET regulatory network involves the transcription factor ERF1, which can regulate the defence gene *PDF1.2* [139]. Vega et al. [102] demonstrated that ERF1 was repressed under N-deficiency in the *Solanum lycopersicum*-*Botrytis cinerea* interaction. A similar repressive effect with N-deficiency was also reported in the *OPR3* gene, which increases tomato resistance to *B. cinerea* by affecting the JA biosynthesis pathway [140]. However, Farjad et al. [141] showed that high N-environment-inhibited JA-related defences, resulted in higher bacterial pathogen cell numbers. Other plant hormones, such as abscisic acid (ABA), auxin, and cytokinin, are involved in plant defence and related to N nutrition. Perhaps most well-characterized is ABA, which can induce stomatal closure to prevent pathogenic bacterial infection but can also negatively affect plant immunity when acting as a signal molecule [142]. Notably, N deficiency can enhance ABA synthesis, while NO_3_^−^ decreases ABA levels when compared with NH_4_^+^ [143,144]. This may represent either nutritional stress or a role for NO and indicates the connection between N metabolism and ABA-regulated plant immunity.

N-mediated defence and homeostasis of phytohormone are also correlated by NO signals, and such interactions have been investigated by Mur et al. [145]. NO is a signal molecule that is closely related to NO_3_^−^ nutrition due to the catalysis of Nr, representing a direct link between N assimilation and NO generation. NO signalling is involved in many facets of plant immunity, such as the transcription regulation of the *PR* gene and local HR-induced PCD [146,147]. NO-regulated PCD at the infection site can arise independently or synergistically with ROS, via the action of antioxidant enzymes or chemical generation of peroxynitrite (ONOO-) [148]. NO signal can also initiate the synthesis of plant phytoalexins or SA with direct roles in chemical defence or indirectly through localized and systemic resistance [149]. Most evidence suggests that the nitrite-dependent Nr pathway is the major source of NO during plant-pathogen interactions [118]. The importance of Nr in plant disease resistance, as well as the correlation with different forms of N, has been highlighted in a number of studies. Thus, NO_3_^−^ nutrition can enhance NO generation via Nr activity, while the NO signal is inhibited under NH_4_^+^ condition [150]. Further, NO_3_^−^ nutrition can regulate NO-mediated responses to pathogen infection, such as stomatal closure or the kinetics of the HR [128]. Thus, the Nr-deficient double mutant (*nia1 nia2)* of *Arabidopsis thaliana* exhibited an impaired HR against *avirulent Pseudomonas* [151]. Fumigation of *nia1nia2* with NO restored the expression of defence-related genes and alleviated the disease symptoms [152]. A similar contribution by the nitrate transporter NRT2.1 has been reported with functions associated with SA and ROS changes [153]. The increased susceptibility of *nrt2.1* to *Erwina amylovora* linked plant defence to redox status and N metabolism, which was suggestive of NO effects. Furthermore, a lower [NH_4_+]/[NO_3_^−^] ratio can act in Arabidopsis *ENHANCED DISEASE SUSCEPTIBILITY 1* (*EDS1*) to augment basal resistance through the action of NO [154]. A study from Gupta et al. [118] demonstrated that the HR-mediated local cell death in response to *Pseudomonas* was faster in NO_3_^−^ fed than in NH_4_^+^ fed tobacco plants, together with the increased plant resistance. Their more recent study associated the nitrite and NO increased hypersensitive response (HR) to the changes of amino acid and energy metabolism [155]. These results positively suggest that the NO signal can closely link N metabolism to plant resistance by manipulating downstream immune responses.

NO can actively participate in the hormone regulatory network, and the associations are demonstrated through the manipulation of haemoglobins (Hb; also known as phytoglobulins) oxidizing NO or a putative plant NO synthase (NOS) catalyzing L-arginine [156,157]. The NO signal can positively impact defence signalling by affecting the expression of the SA biosynthetic gene *isochorismate synthase 1* (*ICS1*) or the JA biosynthesis genes *LOX3*, *12-oxophytodienoate reductase* 1, 2, and 3 [145]. In NO-SA interactions, s-nitrosoglutathione can regulate SA synthesis or plant defence after S-nitrosylation of TGA1 or NPR1 [158]. NO can play either a positive or negative regulatory role in the ETH network. A positive correlation may involve the ETH biosynthetic genes *1-aminocyclopropane-l-carboxylic synthase* (*ACS*) or *1-aminocyclopropane-1-carboxylic acid oxidase* (*ACO*). In contrast, a negative effect could involve suppression of methionine adenosyltransferase activity, which can reduce the provision of methyl groups required for ETH biosynthesis [145].

A transcriptional regulatory network identified by gene network analysis indicated that the hub transcription factor functioned in ETH and JA signalling, which was linked to N-enhanced resistance. These findings suggest the necessity of N in the downstream transcriptional regulation of ETH/JA- mediated defence genes, and similar effects were also reported in *HPL* (encoding hydroperoxide lyase) in melon [159] and antioxidant-related genes in *Populus* [160]. When considering plant susceptibility caused by N nutrition, the interaction between *Magnaporthe oryzae* and rice has been used as a model. Huang et al. [22] revealed that high N fertilization not only promotes the expression of defence genes like *PR* and those involved in the biosynthesis of the chemical defences but also induces several negative regulators of defence during rice-*Magnaporthe oryzae* relations. The study also showed the important role of the *OsNAP* gene in plant immunity, whose expression was eliminated by high N [22]. *OsNAP* has either a negative or positive effect on ABA or JA, which corresponds to the impacts on rice blast resistance [161,162]. A similar observation was noted in the *Arabidopsi*s-*Erwinia amylovora* relations, with up-regulation of multiple resistance genes by N-starvation and pathogen challenge, particularly *PR1*, *WRYK33* and *WRYK60,* well-known positive regulators in plant defence [141]. This may reflect the preservation of plant defence programs under N-starvation stress, while the opposite conclusion cannot be drawn under high N.

Different forms of N ([NH_4_^+^] vs. [NO_3_^−^]) can also impact the plant molecular strategy against pathogens. Generally, NH_4_^+^ mediated plant immunity was associated with the metabolism of amino acids, thereby affecting the development of pathogens or the expression of defence-related genes [163]. A recent study also highlighted the potential negative regulatory role of the ammonium transporter *AMT 1.1* in plant disease resistance, which can be stimulated by N starvation [164]. However, in the context of NO_3_^−^, NO_3_^−^ nutrition inevitably enhances the downstream immune response through the NO signal, thereby improving plant disease resistance.

#### 3.3.2. N Metabolism and SAR

Unlike local defences, SAR has broad-spectrum resistance to all pathogenic microorganisms (e.g., fungi, oomycetes, viruses, and bacteria) and aims to support plant survival rather than death [15]. Therefore, the SAR event is also defined as defence priming [8]. SA is classically associated with SAR, but it has also been linked to the methylated derivative methyl salicylic acid (MeSA), azelaic acid (AzA) and phosphorylated sugar, glycerol-3-phosphate (G3P) [165]. For example, foliar application of SA can stimulate SAR and effectively prevent the development of Fusarium wilt [166]. However, SA is generally not considered to be the main long-range signal in establishing SAR. The establishment of SAR requires contributions from other signalling molecules, such as ROS and RNS, to further induce de novo biosynthesis of SA in systemic leaf tissue [167]. A direct connection between the level of plant in vivo N and the potency of SAR cannot be easily drawn from previous studies. However, it seems likely that N can mediate plant SAR by influencing the production of NO. The generation of a systemic NO signal was first observed in uninoculated tissues above the powdery mildew incubated tomato leaf [167]. A more detailed relationship was demonstrated in *Arabidopsis*-*Pseudomonas syringae* pv *tomato* interactions [168]. Studies demonstrated that NO confers SAR through the downstream regulation of ROS-AzA-G3P, which functioned parallel to SA-mediated events [168]. Recent studies have identified pipecolic acid (Pip), a lysine (Lys) catabolite, as an essential regulator of SAR [169]. Pathogen infection triggers the catabolism of Lys as well as the generation of Pip, which could be transported to the distal tissue and cause an amplification of SA dependent or independent SAR [170]. Pip-mediated SAR has been outlined in multiple plants and the action pattern has been elucidated. Crucially, exogenous application with Pip could reinstate SAR in A*GD2-LIKE DEFENSE RESPONSE PROTEIN1* (*ald1*) mutant further confirmed the relationship between Pip and SAR [72,171]. The role of N-influencing Lys-derived Pip has yet to be investigated but should feature in future studies.

## 4. Conclusions and Perspective

The relationship between N fertilization rates or forms and plant disease incidence is complicated, and the understanding of this relationship or the exploration of the underlying mechanisms is of great significance for agricultural practices. This review summarizes the N regulated effects in the pathogen and host. Its key take-home message must be that no generic model can describe the role of N in a given interaction. Pathogenic mechanisms and host defence responses are too variable. However, some broad themes can be identified, based on which individual exceptions in a particular pathosystem could be subsequently defined. These are described in the host in Figure 2. Firstly, N nutrition influences the strength of plant physical barriers with reduced wax layer thickness and lignin content to affect the penetration by pathogens. Secondly, N-induced biochemical defence represents a network of phytoalexins, antimicrobial proteins, defence-related enzymes, amino acid and organic acid metabolism, and endogenous hormones. Finally, at the molecular level, N nutrition can also manipulate defence-related gene expression via transcriptional regulation or signalling pathways mediated by hormones or amino acids. The NO signal is a crucial link to N forms and also plays an important positive role in both plant local defence and SAR.

Further studies should address the following: (1) the effects of N fertilization forms ([NH_4_^+^] vs. [NO_3_^−^]) on plant thickness of wax layer and cuticle as well as the contents of phytoalexins and antimicrobial proteins when suffering diseases; (2) grafting and gene editing methods can be combined to study the mechanism of N-enhanced (or -compromised) SAR via NO signal or amino acid metabolism; (3) transcriptomics approaches should be broadly used to identify the regulatory pathway between N and HR at local site and SAR at distal site; (4) integrating N effect on the pathogen into the model possibly through approaches such as dual metabolomic approaches, where both host and pathogen are simultaneously assessed.

## Figures and Tables

**Figure 1 ijms-21-00572-f001:**
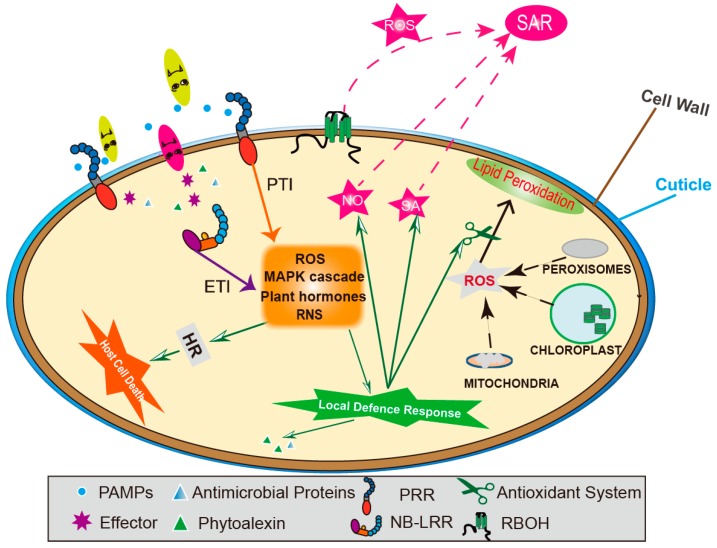
A model of plant immunity to pathogenic microorganism infection. The invading pathogenic microorganisms are first subject to plant physical (the outermost cuticle and epicuticular wax together with the lignified cell wall) and chemical barriers (phytoalexin and antimicrobial proteins). Simultaneously, the pattern recognition receptors (PRRs) located at the membrane recognize pathogen-associated molecular patterns (PAMPs) to induce PAMP-triggered immunity (PTI), while the nucleotide-binding leucine-rich repeat (NB-LRR) proteins recognize effectors delivered by pathogens and induce effector-triggered immunity (ETI). The PTI and ETI activate downstream defence including the local defence response and host cell death, mediated by a series of signal or regulatory factors (such as reactive oxygen species (ROS), reactive nitrogen species (RNS), mitogen-activated protein kinase (MAPK) cascades and hormones). The antioxidant systems are also stimulated to maintain intracellular redox balance. Furthermore, signals including nitric oxide (NO), salicylic acid (SA) and NADPH oxidase (RBOH)-generated ROS act to induce systemic acquired resistance (SAR) in uninfected tissues.

**Figure 2 ijms-21-00572-f002:**
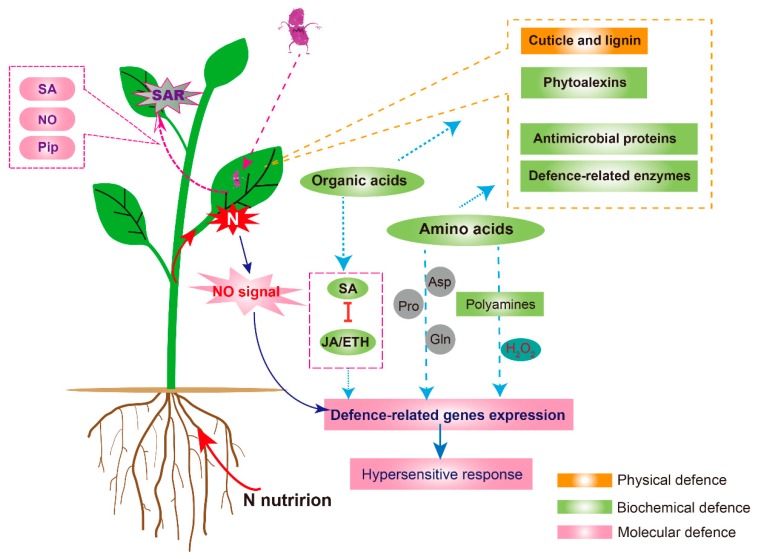
Proposed model for the interaction between nitrogen (N) nutrition and plant physical, biochemistry and molecular defences. N nutrition is involved in the construction of the plant defence system, including the defined negative effects on physical defences and phytoalexins contents and the positive effects on antimicrobial proteins. N also mediates phytoalexin formation and downstream hypersensitive response via the amino acid metabolism. In addition, N nutrition can also regulate hormone (salicylic acid, SA; jasmonic acid, JA and ethylene, ETH) status and the subsequent defence genes expression patterns through organic acid metabolism or the function of nitric oxide (NO). NO signal is an important bridge between N nutrition and plant disease resistance, which is tightly linked to N forms. Notably, pipecolic acid (Pip) derived from lysine (Lys), NO signal, and SA can also mediate the distant systemic acquired resistance (SAR) together with reactive oxygen species (ROS), which is critical for plant survival when facing diseases.

**Table 1 ijms-21-00572-t001:** Number of published papers reporting the effects of nitrogen nutrition on plant disease incidence.

Disease Incidence	Effect of Nitrogen in the Form of		
	Unspecified N	NH_4_^+^	NO_3_^−^
Cases	73	19	22
Increase in disease	40	9	13
Decrease in disease	25	9	8
No effect or variable	8	1	1

The data were collected from 132 published papers that related to nitrogen nutrition and plant disease ranging from 1944 to 2019.

**Table 2 ijms-21-00572-t002:** Defence-related enzymes or compounds regulated by nitrogen during plant disease resistance.

Host	Disease	Pathogen	Defence-Related Enzymes/Compounds	Reference
Apple tree	Scab disease	*Venturia inaequalis*	Procyanidins, Flavonols	[86]
Grapevines	Powdery mildew	*Uncinula necator*	Flavonol glycosides, Cinnamic acid	[83]
Medicago truncatula	Root rot	*Aphanomyces euteiches*	Soluble phenolics, Phenylalanine ammonia lyase	[84]
Potato	Early blight	*Alternaria solani*	Chlorogenic acid, Flavonols, Neochlorogenic acid	[85]
	Late blight	*Phytophthora infestans*	Phenylalanine ammonia lyase, Polyphenol oxidase, chitinase, Flavonols	[93]
	Leaf necrosis	*potato virus Y*	Phenylalanine ammonia lyase	[94]
Rice	Rice blast	*Magnaporthe grisea*	Sulfur-rich thaumatin-like protein	[95]
		*Magnaporthe oryzae*	Phenylalanine ammonia lyase, Superoxide dismutase, Glucanases, Chitosanase, Phenylalanine ammonia lyase,	[24,96]
	Sheath rot	*Sarocladium oryzae*	Phenols	[82]
Tomato	Bacterial speck	*Pseudomonas syringae*	Superoxide dismutase	[97]
	Fusarium wilt	*Fusarium oxysporum*	Phenols, peroxidase	[98]

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
