# Peer review of "Unravelling the Roles of Nitrogen Nutrition in Plant Disease Defences"

_ijms, 2020, doi:10.3390/ijms21020572_

Round 1
Reviewer 1 Report
Dear Author,
The manuscript is the compilation of paper based on the role of N in plant disease defense.
Also, its negative effects on physical defense and the production of anti-microbial phytoalexins but positive effects on defense-related enzymes and proteins.
Manuscript well organised and describe with supported figures and tables.
Author Response
Point 1: The manuscript is the compilation of paper based on the role of N in plant disease defense. Also, its negative effects on physical defense and the production of anti-microbial phytoalexins but positive effects on defense-related enzymes and proteins. Manuscript well organised and describe with supported figures and tables. 

Response 1: Thank you very much for your affirmation and encouragement concerning our manuscript. In the revised version, we double-checked and changed the incorrect punctuations and the format of the references all through the manuscript. We also carefully rechecked English writing to avoid any possible mistakes. Thank you again for your encouragement.
Reviewer 2 Report
The paper titled ¨Unravelling the roles of nitrogen nutrition in plant disease defences¨ provides useful information about how nitrogen nutrtion affects plant defense and which mechanisms are involved.However, there are some concerns regarding the paper that should be further resolved before its publication.
some sections of the review (abstract, introduction, N nutrition as it impacts on the pathogen, N nutrition as it impacts on host defence)are difficult to read and follow. To make it easier for the readers, the review should be checked by a professional english reader.
the introduction provides a general description of plant defenses. However, the authors only mention SAR when they talk about induced resistance. The other forms of induced resistance should be mentioned
table 2 is mentioned in the text in section 3.2.2, however it also provides examples of other defense metabolites. please include it into the text accordingly
information related to N nutrition influence on ROS should be added
in section 3.3.1 information about the influence of N nutrition on other hormones (for example: ABA) should be added
Author Response
Point 1: The paper titled ¨Unravelling the roles of nitrogen nutrition in plant disease defences¨ provides useful information about how nitrogen nutrtion affects plant defense and which mechanisms are involved.However, there are some concerns regarding the paper that should be further resolved before its publication.
Response 1: Thank you very much for your encouragement and for your valuable suggestions and comments concerning our manuscript. We have considered all comments carefully and revised our manuscript, especially for the English writing, which we hope will meet with approval for publication. Besides, we also changed the incorrect punctuations and the format of the references all through the manuscript to avoid any possible mistakes. Please check the new vision.
Point 2: some sections of the review (abstract, introduction, N nutrition as it impacts on the pathogen, N nutrition as it impacts on host defence) are difficult to read and follow. To make it easier for the readers, the review should be checked by a professional english reader. 

Response 2: Sorry for our careless and thank you very much for your comments on English writing, which is valuable for improving our entire manuscript. In the revised version, we have carefully checked and corrected all errors in English writing to avoid any possible mistakes. Furthermore, the writing of the entire manuscript was improved by Prof. Luis Mur, a native English speaker.
Point 3: the introduction provides a general description of plant defenses. However, the authors only mention SAR when they talk about induced resistance. The other forms of induced resistance should be mentioned
Response 3: Thank you very much for your suggestion. We added other forms of induced resistance in the revised version, such as the systemic acquired resistance triggered by pathogens, systemic resistance caused by beneficial microbes including plant-growth-promoting rhizobacteria and fungi, resistance induced by herbivore and resistance stimulated by chemical inducer or abiotic stress. Please check the revised version on page 2-3, lines 72-75.
References:
Mauch-Mani, B.; Baccelli, I.; Luna, E.; Flors, V. Defense priming: an adaptive part of induced resistance. Annu. Rew. Plant Biol. 2017, 68, 485-512.
Point 4: table 2 is mentioned in the text in section 3.2.2, however it also provides examples of other defense metabolites. please include it into the text accordingly
Response 4: Thank you very much for your suggestions. We have included Table 2 into the text related to other defense metabolites, including the phenolic compounds, antimicrobial proteins, defence-related enzymes and antioxidant systems. We also adjusted the position of Table 2 to the paragraph where the content was first mentioned. Please check the revised version, lines 203, 225, 292 and 310. The changes in the references due to the change in the position of Table 2 have also been marked; please check the references section.
Point 5: information related to N nutrition influence on ROS should be added
Response 5: Thank you very much for your suggestion. We have added information related to the influence of N nutrition on ROS levels in the revised version. N nutrition can regulate the antioxidant system and therefore impact the ROS accumulations. Studies have demonstrated that N-deficient would increase the accumulations of hydrogen peroxide and superoxide anion, which is consistent with the enhanced antioxidant system under high-N condition. Besides, N supply forms would also affect ROS levels. The NO3− feeding caused decrease of ROS levels may be linked to the nitric oxide signal. Please check the revised version on page 8, lines 314-320.
References:
Lin, Y.; Chao, Y.; Huang, W.; Kao, C. Effect of nitrogen deficiency on antioxidant status and Cd toxicity in rice seedlings. Plant Growth Regul. 2011, 64, 263-273.
Kovacik, J.; Klejdus, B.; Backor, M. Nitric oxide signals ROS scavenger-mediated enhancement of PAL activity in nitrogen-deficient Matricaria chamomilla roots: side effects of scavengers. Free Radical Bio. Med. 2009, 46, 1686-93.
Kong, L.; Wang, F.; Si, J.; Feng, B.; Zhang, B.; Li, S.; Wang, Z. Increasing in ROS levels and callose deposition in peduncle vascular bundles of wheat (Triticum aestivum L.) grown under nitrogen deficiency. J. Plant Interact. 2013, 8, 109-116.
Ali, S.; Farooq, M.A.; Jahangir, M.M.; Abbas, F.; Bharwana, S.A.; Zhang, G.P. Effect of chromium and nitrogen form on photosynthesis and anti-oxidative system in barley. Biol. Plantarum 2013, 57, 758-763.
Point 6: in section 3.3.1 information about the influence of N nutrition on other hormones (for example: ABA) should be added
Response 6: Thank you very much for your comments and valuable suggestions. We have added information about the influence of N nutrition on other hormones in section 3.3.1. salicylic acid, jasmonic acid and ethylene are major plant defence regulators, while other hormones, such as abscisic acid (ABA), auxin, and cytokinin, are also involved in plant defence. ABA is the most characterized one among these hormones, which can either positively regulate plant defence against pathogenic bacterial infection via stomatal regulation, or negatively regulate plant immunity when acting as a signal molecule. Crucially, N deficiency can enhance ABA synthesis, while NO3− would decrease ABA levels when compared with NH4+, which may represent either nutritional stress or a role for NO. Please check the revised version on page 9, lines 341-347.
References:
Berens, M.L.; Berry, H.M.; Mine, A.; Argueso, C.T.; Tsuda, K. Evolution of hormone signaling networks in plant defense. Annu. Rew. Phytopathol. 2017, 55, 401-425.
Oka, M.; Shimoda, Y.; Sato, N.; Inoue, J.; Yamazaki, T.; Shimomura, N.; Fujiyama, H. Abscisic acid substantially inhibits senescence of cucumber plants (Cucumis sativus) grown under low nitrogen conditions. J. Plant Physiol. 2012, 169, 789-796.
Garnica, M.; Houdusse, F.; Zamarreno, A.M.; Garcia-Mina, J.M. The signal effect of nitrate supply enhances active forms of cytokinins and indole acetic content and reduces abscisic acid in wheat plants grown with ammonium. J. Plant Physiol. 2010, 167, 1264-72.